# Exploring the Link Between Suicidal Concern and Pharmacotherapy in Adolescents: Evidence from a Clinical Cohort

**DOI:** 10.3390/jcm14186412

**Published:** 2025-09-11

**Authors:** Francesca Marazzi, Marika Orlandi, Arianna Vecchio, Valentina De Giorgis, Martina Maria Mensi

**Affiliations:** 1Department of Brain and Behavioral Sciences, University of Pavia, 27100 Pavia, Italyvalentina.degiorgis@mondino.it (V.D.G.);; 2Child Neurology and Psychiatry Unit, IRCCS Mondino Foundation, 27100 Pavia, Italy

**Keywords:** suicide, adolescence, psychopharmacological treatment, suicidal risk, medications, prescriptions

## Abstract

**Background/Objectives:** Suicidal risk is a major public health concern among adolescents. Pharmacological treatment in this population remains complex and often targets underlying psychiatric disorders rather than suicidal risk itself. This study aimed to examine associations between suicidal concern (SC) and psychotropic prescriptions in hospitalized adolescents with psychiatric disorders. A secondary aim was to assess whether suicidal risk level predicted pharmacological treatment at discharge (T1). **Methods:** A cross-sectional study was conducted on 224 adolescents (men age = 15.4, SD = 1.55). SC was assessed using the Columbia Suicide Severity Rating Scale (C-SSRS). Participants were categorized into SC and without SC (No Suicidal Concern—NSC) groups and further stratified by low and high suicidal risk. Psychotropic prescriptions at admission (T0) and T1 were compared, adjusting for age, gender, and psychiatric diagnoses. **Results:** The SC group showed more severe clinical presentations, including higher symptom burden and functional impairment. At discharge, they were more frequently prescribed antipsychotics and supplements, and more often received multiple medications compared to the NSC group. Logistic regression confirmed SC as a significant predictor of antipsychotic and supplement prescriptions at T1, independent of age, gender, and diagnosis. In contrast, suicidal risk level did not significantly predict specific prescriptions or polypharmacy prescriptions. **Conclusions:** SC appears to influence pharmacological decisions beyond diagnostic classifications, with a tendency toward risk-containment strategies. These findings emphasize the need for individualized, developmentally appropriate, and evidence-based treatment planning. Importantly, SC should be considered not only as a symptom but as a potential independent treatment target in adolescent psychiatry.

## 1. Introduction

Suicide is among the leading causes of death worldwide and represents a crucial global public health concern, accounting for more than 700,000 deaths each year [1]. Among adolescents and young adults aged 15 to 29 years, it ranks as the third leading cause of death [2]. For years, the World Health Organization (WHO) has recommended that suicide prevention be treated as a public health priority and has called for the development and implementation of comprehensive national strategies, with particular attention to youth and other vulnerable groups [3]. Suicidal thoughts and behaviors frequently emerge during the transition from childhood to adolescence [4]. Concurrently, the literature highlights that individuals who engage in self-injurious attempts at a young age are more likely to experience psychiatric disorders throughout their lives [5].

In adolescents, the presence of comorbid psychopathological conditions plays a critical role [6], representing a significant risk factor, often with onset during adolescence [7]. Indeed, the literature documents a strong association between suicidal concern in youth and mental disorders, particularly mood, anxiety, behavioral, attention disorders, and substance use disorders [2,8]. A previous suicide attempt is one of the most robust predictors of suicidal risk during adolescence and across the lifespan [9,10]. Additionally, impulsivity [11], emotional dysregulation [12,13,14], and hopelessness [15] have been identified as predictors of suicidal thoughts and/or suicidal behaviors (SBs).

Beyond psychopathological conditions, environmental and social factors also contribute to the development of suicidal risk, such as dysfunctional family dynamics [16] and a lack of psychosocial support [17]. However, due to the multifactorial and complex nature of suicidal risk, individual risk factors show limited predictive power. For instance, although psychiatric disorders are among the most extensively studied predictors [18,19], they do not fully account for the onset of suicidal ideation (SI) [9]. Accordingly, the scientific community is increasingly called upon to explore transdiagnostic factors that may contribute to the emergence of suicidal risk either independently or through psychiatric comorbidity [20].

The guidelines do not suggest a single possible treatment, but rather a combination [2,21,22]. Focusing on medication treatments, among the most studied medications for treating SI and SBs in adults are lithium, ketamine, and clozapine [23,24], although studies in pediatric populations remain limited [25,26]. Pharmacological treatments for suicidal concern in children and adolescents are typically off-label and primarily target underlying psychiatric conditions rather than suicidal concern itself [24,25]. The heterogeneous and multifactorial nature of suicidal thoughts and behaviors complicates the selection of appropriate pharmacological therapy, particularly in childhood and adolescence.

Thus, the role of pharmacological interventions in managing suicidal concern during childhood and adolescence appears complex. The various drug classes used to treat psychiatric conditions (such as anxiety, depression, mania, and psychosis) are not explicitly recommended for the prevention or treatment of SI or SB [27]. Nevertheless, the treatment of underlying psychiatric conditions associated with suicide risk has been recognized as a key component in the management of SB and suicide prevention [24].

In light of that, the first aim of this study was to examine the association between the presence of any suicidal concern (SC) and psychiatric pharmacological prescriptions in a clinical sample of adolescents affected by psychiatric disorders. Participants were categorized into two groups based on the presence (concern) or absence (no concern) of SI and/or SB, as assessed through the Columbia Suicide Severity Rating Scale (C-SSRS) [28]. We compared pharmacological prescriptions between these groups at admission (T0) and discharge (T1). Also, we examined whether the presence of suicidal concern was a significant predictor of prescription at T1, adjusting for age, gender, and psychiatric diagnoses.

The second aim of the study was to explore whether adolescents classified as high-risk for suicidal ideation and/or behaviors, based on the C-SSRS, were more likely to be prescribed psychiatric medications compared to those at low risk. We hypothesized that patients at higher clinical risk would receive more frequent prescriptions of pharmacological treatments at T1, independently of psychiatric diagnosis, age, and gender.

## 2. Materials and Methods

### 2.1. Study Design

This cross-sectional cohort study received approval from the Ethics Committee of Policlinico San Matteo in Pavia (P-20170016006). The Committee reviewed the study protocol, informed consent and assent forms, and other relevant documentation, discussed the project within a multidisciplinary panel, and issued its opinion in accordance with standard procedures. The study was conducted following the REporting of studies Conducted using Observational Routinely collected health Data (RECORD) statement (Appendix A). The authors assert that all procedures contributing to this work comply with the ethical standards of the relevant national and institutional committees on human experimentation and with the Declaration of Helsinki of 1964 and its subsequent amendments [29]. All patients and their caregivers provided written informed consent to participate in the study and were free to withdraw at any moment without explanation. The data were pseudonymized and are available in the Zenodo repository upon request [30].

### 2.2. Participants and Procedures

Of the 350 adolescents, aged between 12 and 18 years, admitted to the Child Neurology and Psychiatry Unit of the tertiary care IRCCS Mondino Foundation (Pavia, Italy) between May 2020 and July 2025, 126 were excluded, resulting in a final study sample of 224 participants (mean age = 15.4, SD = 1.55, 186 females).

Diagnoses were made according to the Diagnostic and Statistical Manual of Mental Disorders, Fifth Edition (DSM-5) criteria and the Diagnostic and Statistical Manual of Mental Disorders, Fifth Edition, Text Revision (DSM-5-TR) [31,32].

We excluded individuals with: (a) intellectual disability (IQ ≤ 70) evaluated using the age-appropriate Wechsler intelligence scale (WISC-IV, WISC-V, or WAIS-IV) [33,34,35]; (b) inadequate proficiency in the Italian language; (c) a history of significant head trauma or the presence of any medical or neurological condition that could affect participation. Exclusions were entirely due to a priori criteria rather than random attrition. Figure 1 shows a detailed study population flowchart.

All adolescents involved in the study underwent a comprehensive psychodiagnostic assessment, which allowed for the formulation of a psychiatric diagnosis according to DSM-5 and DSM-5-TR criteria [31,32]. Among the instruments used to establish the diagnosis, we included the Kiddie Schedule for Affective Disorders and Schizophrenia, Present and Lifetime Version (K-SADS-PL-DSM-5) [36], a semi-structured interview based on DSM-5 criteria that explores current and past psychopathology and comorbidities (e.g., mood disorders, anxiety disorders, OCD, nutrition disorders, psychotic disorders). The K-SADS-PL has demonstrated good interrater reliability and validity across languages [37]. For participants aged 14 years and older, problematic personality traits or personality disorders were assessed using the Structured Clinical Interview for DSM-5 Personality Disorders (SCID-5-PD) [38], a semi-structured diagnostic interview with well-established validity and reliability in several languages [39,40]. For each patient, clinicians also completed the Clinical Global Impression-Severity scale (CGI-S) [41], a valid and reliable [42] clinician-report measure designed to capture symptoms’ overall intensity and severity on a 7-point scale from 1 (normal, not at all ill) to 7 (among the most extremely ill patients), compared to the clinician’s experience with similar patients. Clinicians also completed the Children’s Global Assessment Scale (CGAS) [43], a clinician-report scale that provides standardized ratings of young patients’ overall functioning in social and occupational domains on a 100-point rating scale ranging from 0 (highly impaired) to 100 (excellent functioning).

All participants were administered the semi-structured interview C-SSRS [28], an assessment tool that, through dichotomous responses (yes/no), evaluates suicidal severity and intensity within the 6 months before the interview and over the individual’s lifetime.

A score of 1 represents a wish to be dead, 2 corresponds to non-specific active suicidal thoughts, 3 signifies active suicidal ideation with any method (without a specific plan) and without intent to act, 4 denotes active suicidal ideation with some intent to act but without a specific plan, and 5 reflects active suicidal ideation with a detailed plan and intention to die. This semi-structured interview also assesses previous and actual history of suicidal behaviors (actual, interrupted, and aborted suicide attempts over the lifetime) using a dichotomous response format, along with the number of each type of attempt and any preparatory acts.

Participants were classified into two groups according to C-SSRS scores. The No Suicidal Concern (NSC) group included patients who received a psychiatric diagnosis according to DSM-5 or DSM-5-TR criteria, but who did not report any SI and/or SB (negative responses to all items of the C-SSRS). In contrast, the Suicidal Concern (SC) group included patients with a psychiatric diagnosis who endorsed SI and/or SB (at least one positive response on the C-SSRS).

Moreover, we formulated two suicidal risk levels for the participants in the SC group: low and high risk. The low-risk group included adolescents with passive SI (positive responses to C-SSRS items 1–3 in the past 6 months) and no positive responses concerning SBs. The high-risk group included those who endorsed at least one of the following: active suicidal ideation with intent or plan (C-SSRS items 4 or 5), and/or any SB (concrete, interrupted, aborted attempts, or preparatory acts).

In addition, clinicians collected detailed sociodemographic data for each patient, including social relations, school performance, and risky behaviors. These variables were systematically coded at T0 based on clinical interviews and available documentation. Social relations were rated as social withdrawal, poor, or adequate, reflecting the adolescent’s overall ability to establish and maintain interpersonal relationships. School performance was categorized as poor, sufficient, good, excellent, or withdrawal, based on teachers’ reports, parental information, and academic records, where available. Risky behaviors included non-suicidal self-injury (e.g., cutting), drug use, suicidal behaviors (as assessed through the C-SSRS), and combinations of these behaviors, with detailed categories reported in Table 1.

Furthermore, a comprehensive pharmacological history was obtained at T0, and any modifications or additions to pharmacological treatment during hospitalization were systematically documented. Thus, we reported prescriptions at discharge (T1). Hospitalization usually lasted from 7 to 15 days.

### 2.3. Data Analysis

Analyses were conducted using RStudio (v2025.5.0.496) [44]. Descriptive statistics were computed to summarize sociodemographic, clinical, and pharmacological variables, and were presented as means and standard deviations (SD) for continuous variables, and frequencies and percentages for categorical variables. Effect sizes were reported where applicable (Cohen’s d, Cramer’s V).

Group comparisons between patients with and without suicidal concern (SC vs. NSC) and between low- and high-risk levels within the SC group were performed using Fisher’s exact test (when expected cell counts were small) or chi-square test for categorical variables, and Welch’s *t*-test for continuous variables, given its robustness to unequal variances.

Binary logistic regression analyses were conducted to examine the association between pharmacological prescriptions at T1 and suicidal concern (Aim 1), and between prescriptions and suicidal risk level (low vs. high) within the SC group (Aim 2). All models were adjusted for age, gender, and psychiatric diagnoses, given their known associations with both suicidal risk and psychiatric medication use.

McNemar’s test was used to assess within-group changes in prescription patterns from T0 to T1, separately for SC/NSC and low/high risk groups, as it is designed for paired categorical data. Analyses were performed on complete cases (listwise deletion), with no imputation of missing values.

Furthermore, an exploratory analysis of polypharmacy (i.e., number of drug classes prescribed at T1) was conducted using Welch’s *t*-test to compare low- and high-risk groups.

Finally, to assess the robustness of the findings, we performed sensitivity analyses restricted to two diagnostic subgroups: patients with depressive disorder, and patients with depressive and/or anxiety disorder. We repeated group comparisons, McNemar’s tests, and multivariate logistic regressions, following the same analytical approach applied in the primary analyses.

Statistical significance was set at *p* < 0.05 (two-tailed).

## 3. Results

The study sample consisted of 224 adolescents diagnosed with at least one psychiatric disorder. Table 1 shows the sociodemographic, clinical, and pharmacological characteristics of the total sample and the two subgroups. The SC and NSC groups statistically differ in gender, social relations, and risky behaviors. Concerning diagnoses, they differ in psychotic, depressive, and anxiety symptoms, and personality disorder traits, as well as syndrome severity, and global functioning. Moreover, the SC group shows more antipsychotic prescriptions than the NSC group and a higher number of medications at T1.

### 3.1. Aim 1: Association Between Suicidal Concern and Pharmacological Prescriptions

Group comparisons between SC adolescents and NSC ones showed statistically significant differences in several psychotropic prescriptions at T1, as shown in Table 2. Antipsychotics and benzodiazepines were prescribed more frequently in the SC group compared to the NSC, as well as supplements. Mood stabilizers were used rarely overall but more frequently in the SC group. Receiving any pharmacological treatment was more frequent in the SC group than in the NSC. No significant group differences were found for antidepressants.

McNemar’s tests were used to assess changes in prescriptions within each group. Among SC patients, antidepressants, antipsychotics, and supplements increased from T0 to T1. Total pharmacological treatments significantly increased. In the NSC group, only antidepressant prescriptions showed a significant increase. No significant changes were observed for other medication classes, as shown in Table 3.

A series of binary logistic regression models was conducted to assess the association between pharmacological prescriptions at T1 and SC, adjusting for age, gender, and psychiatric diagnoses. Significant associations were found for antipsychotic prescriptions, supplement prescriptions, and total drug prescriptions. No significant associations were observed for antidepressants, benzodiazepines, or mood stabilizers (results inconclusive due to data issues) (Table 4).

### 3.2. Aim 2: Association Between Suicidal Risk Level and Pharmacological Prescriptions

Within the SC group, 30 patients were classified as low-risk and 116 as high-risk based on the C-SSRS classification. As shown in Appendix A, the two subgroups did not significantly differ in most sociodemographic characteristics (e.g., age, gender, ethnicity) or psychiatric diagnoses. The only clinical variables that showed meaningful differences were risky behaviors, more common in the high-risk group, and eating disorders, more frequent in the low-risk group.

McNemar tests were used to explore changes in prescriptions from T0 to T1, stratified by suicidal risk level. In the high-risk group, prescriptions significantly increased for antidepressants, antipsychotics, and supplements. Additionally, the total number of prescriptions increased for the high-risk group from T0 to T1. No significant changes were found in the low-risk group (Table 5).

Binary logistic regression analyses did not reveal any statistically significant association between suicidal risk level and pharmacological prescriptions at T1, when adjusting for age, gender, and psychiatric diagnosis. Odds ratios were above 1 for all drug classes, suggesting a trend toward a greater likelihood of prescription in high-risk patients. However, the confidence intervals were wide, and the *p*-values were non-significant.

### 3.3. Exploratory Analysis: Number of Pharmacological Prescriptions (Polypharmacy)

We investigated whether the number of pharmacological prescriptions at T1 differed between patients classified as having low versus high risk of SI and SBs. On average, high-risk patients received slightly more prescriptions (M = 2.01, SD = 1.28, range: 0–5) compared to low-risk patients (M = 1.7, SD = 1.15, range: 0–4). However, the difference did not reach statistical significance in a Welch’s *t*-test (t = –1.28, df = 49.15, *p* = 0.206; 95% CI [–0.79, 0.18]). This suggests a non-significant trend toward higher polypharmacy in the high-risk group, warranting further investigation in larger samples.

### 3.4. Sensitivity Analyses

In the depression-only subgroup, Fisher’s tests did not reveal significant differences in prescriptions at T1 between SC and NSC patients (Appendix A). By contrast, in the depression/anxiety subgroup, SC adolescents received significantly more antipsychotics and total medications (Appendix A).

McNemar tests showed that, in the depression-only group, significant within-group increases from T0 to T1 were observed for total prescriptions among SC adolescents (Appendix A). In the depression/anxiety subgroup, SC patients showed significant increases in antidepressant, supplement, and total prescriptions (Appendix A).

Logistic regression confirmed that, in the depression-only group, no pharmacological prescriptions were significantly associated with SC (Appendix A). In the depression/anxiety group, antipsychotic prescriptions were significantly associated with SC (OR = 5.25, *p* = 0.033), while total prescriptions approached significance (OR = 4.35, *p* = 0.052), supporting the robustness of the main findings across diagnostic subgroups (Appendix A).

## 4. Discussion

This study investigated the relationship between SC, defined as the presence of SI and/or SB, and psychiatric pharmacological prescriptions in a clinical sample of adolescents affected by psychiatric disorders. We explored both cross-sectional differences and longitudinal changes in pharmacological prescriptions at T0 and T1, adjusting for age, gender, and psychiatric diagnoses.

At baseline, the SC group displayed significantly greater clinical severity, including more frequent engagement in risky behaviors and lower overall functioning compared to the NSC group, indicating a more complex and acute clinical profile. These findings align with the previous literature describing SC as a marker of psychiatric severity in youth [45].

These differences were reflected in pharmacological patterns at T1, with the SC group receiving a broader range of psychotropic prescriptions, particularly antipsychotics and supplements, and were more likely to receive multiple medications overall than their counterparts. This indicates that suicidal concern may influence pharmacological decision-making, independently of diagnostic criteria, potentially favoring medications perceived as rapidly effective in managing behavioral dysregulation or acute risk.

While benzodiazepines were more frequently prescribed in the SC group, the adjusted logistic regression did not show a statistically significant association, suggesting that their prescription may reflect clinical trends rather than a systematic response to SC. Mood stabilizers were rarely prescribed and exclusively within the SC group, highlighting the challenges of studying less common prescriptions.

When examining prescription changes over time, we observed a general increase in pharmacological treatments from T0 to T1, especially for the SC group. Antidepressants, antipsychotics, and supplements were more frequently prescribed at T1, while in the NSC group, only antidepressants showed a significant increase. Furthermore, logistic regression indicated that SC status significantly predicted antipsychotic and supplement prescriptions at T1, underscoring the potential role of suicidal concern as a driver of treatment escalation.

As a second aim, we explored whether suicidal risk level was associated with pharmacological treatment, regardless of diagnosis and demographics. As expected, adolescents in the high-risk group showed significant increases in prescriptions for antidepressants, antipsychotics, and supplements from T0 to T1. However, risk level did not significantly predict pharmacological treatment when controlling for demographics and diagnosis, nor was the overall number of medications at T1 associated with risk level. This suggests that the presence of SC, rather than its severity, may be the primary factor influencing clinicians’ pharmacological decisions.

These findings suggest that treatment strategies for adolescents are often driven by the need to contain acute risk and manage crisis-level behaviors, rather than by strictly diagnostic frameworks. While acute management and immediate safety are necessary priorities in inpatient settings [46], this raises concerns about the long-term appropriateness, coherence, efficacy, and safety of specific pharmacological treatments. For example, the frequent use of antipsychotics in adolescents at risk of suicide may reflect attempts to stabilize crises but warrants critical examination in light of potential side effects and off-label usage [47,48].

Psychopharmacological interventions in adolescents are often adapted from adult models, despite the lack of robust evidence supporting their efficacy and tolerability in younger populations (e.g., lithium, esketamine) [23,24]. As emphasized by previous studies [24], most psychotropic drugs prescribed in adolescence are used off-label, with few age-specific randomized controlled trials to guide treatment [49]. A particularly controversial case is that of antidepressants. In 2004, the U.S. Food and Drug Administration (FDA) issued the so-called Black Box Warning, based on studies highlighting an increased risk of suicidal ideation and behaviors in children, adolescents, and young adults during the initial period of treatment [50,51,52,53]. This warning has since underscored the importance of cautious prescribing of antidepressants in these age groups. Nevertheless, it is crucial to weigh this risk against the high burden of untreated depression and anxiety disorders in youth, which are themselves strongly associated with increased suicide risk and represent a major public health concern [52,54]. In this context, our findings, and those from other studies [55], suggest a consistent rise in antidepressant use in suicidal youth, which may indicate a potential divergence between clinical guidelines and real-world prescribing practices.

This tendency to favor pharmacological strategies in high-risk situations, often beyond strict diagnostic indications, underscores the need for caution. Clinicians may be driven by the urgency to mitigate acute risk, yet the long-term outcomes of such practices remain largely underexplored. There is a risk that treatment choices aimed at short-term behavioral control may not sufficiently address the underlying psychopathological processes and may even reinforce a symptom-suppression model that neglects developmental, relational, and functional dimensions.

In light of these challenges, several recent contributions [22,23,56] have highlighted the importance of a developmentally sensitive and multidimensional approach to suicidal risk in adolescence. They proposed a paradigm shift: rather than treating SC as a byproduct of broader psychiatric disorders, it should be approached as an independent clinical target, warranting direct and sustained therapeutic attention. This perspective, echoed in emerging treatment guidelines, calls for a more personalized, dimensional, and transdiagnostic approach to suicide risk in youth. Hence, pharmacological decisions should be guided not only by formal diagnosis, but also by individual functioning, psychosocial context, and behavioral risk factors. Understanding the associations between suicidal concern and specific drug classes could help clarify the implicit clinical heuristics underlying real-world treatment decisions, ultimately guiding more personalized and evidence-based interventions [57].

Ultimately, future research should assess whether viewing suicidal risk as a distinct therapeutic focus, rather than a secondary transdiagnostic symptom, can improve clinical outcomes and safety in this vulnerable population.

This study should be interpreted considering limitations. The sample size was relatively small and drawn from a single tertiary care center that predominantly treats adolescents with severe and complex psychopathological profiles, limiting generalizability. Second, the sample was predominantly female, reflecting the well-documented higher prevalence of SC and mental disorders among adolescent females [58] but reducing applicability to male populations, who are less likely to seek help but have higher suicide mortality [17,59,60]. Additionally, the present study examined the differences between groups with and without SC. However, further research could investigate similar differences in the pharmacological domain among patient groups characterized by SI, suicide attempts, and non-suicidal ideation and/or attempts. Moreover, heterogeneity in pharmacological treatment regimens (dosage, polypharmacy) could not be controlled, and the presence of other treatments such as psychotherapy was not explored. Finally, the cross-sectional design of this study precludes any causal interpretations. Future studies should aim to include larger, gender-balanced samples, as well as different severities of psychopathology.

## 5. Conclusions

This study suggests that SC significantly influences pharmacological decisions in adolescent psychiatric care, often prompting prescriptions aimed at short-term risk containment, particularly antipsychotics and supplements, as well as an overall greater number of medications. Such strategies may not always align with developmental needs or international guidelines, particularly given the limited evidence for their efficacy and safety in youth. The findings underscore the importance of moving beyond diagnosis-based prescribing and considering SC as an independent treatment target. Clinicians are encouraged to adopt individualized, developmentally informed approaches that integrate psychosocial and functional assessments when planning interventions. Optimizing care for at-risk adolescents requires balancing immediate safety with long-term therapeutic goals.

## Figures and Tables

**Figure 1 jcm-14-06412-f001:**
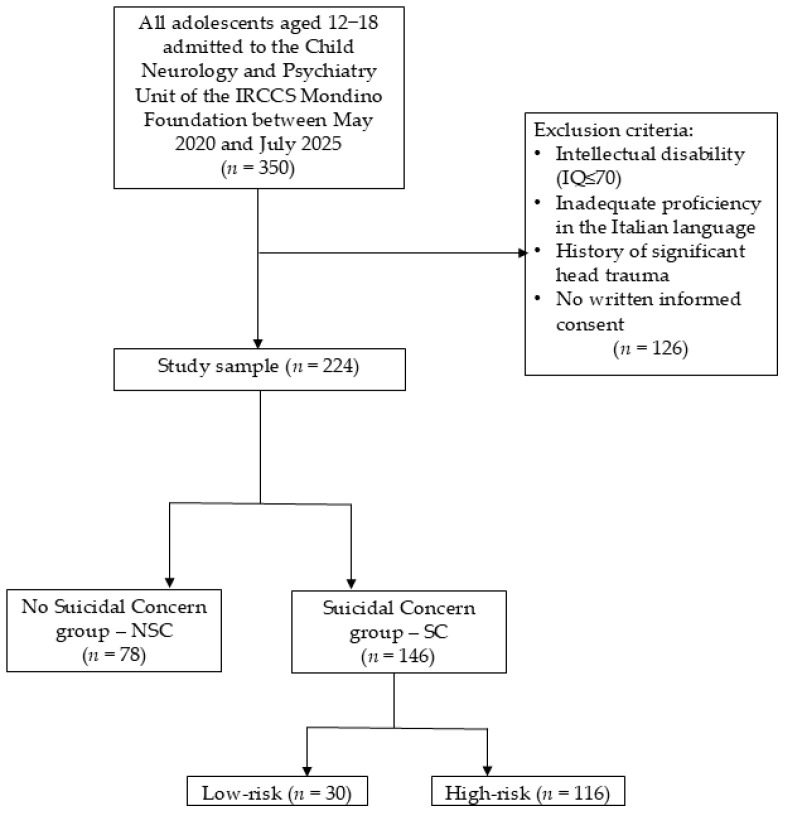
Study population flowchart [authors’ own processing].

**Table 1 jcm-14-06412-t001:** Sociodemographic, diagnostic, and pharmacological data [authors’ own processing].

Variable	Total (*n* = 224)	NSC (*n* = 78)	SC (*n* = 146)	*p*	Effect Size
Females *n* (%)	186 (83%)	58 (73.8%)	128 (88.2%)	0.019 *	0.156
Males *n* (%)	38 (17%)	20 (25.6%)	18 (12.3%)	0.019 *	0.156
Age M ± SD	15.4 ± 1.55	15.22 ± 1.59	15.5 ± 1.52	0.209	−0.179
Ethnicity				0.791	0.123
Caucasian	198	70 (89.7%)	128 (88.8%)		
Asian	5	2 (2.6%)	3 (2.1%)		
African	6	1 (1.3%)	5 (3.5%)		
Latina	7	3 (3.8%)	4 (2.8%)		
Mixed	6	2 (2.6%)	4 (2.8%)		
Social relations				0.005 **	0.218
Social withdrawal	22	5 (6.5%)	17 (11.9%)		
Poor	99	26 (33.8%)	73 (51%)		
Adequate	99	46 (59.7%)	53 (37.1%)		
School performance				0.351	0.142
Poor	28	7 (9.1%)	21 (14.6%)		
Sufficient	57	19 (24.7%)	38 (26.4%)		
Good	87	37 (48.1%)	50 (34.7%)		
Excellent	26	7 (9.1%)	19 (13.2%)		
Withdrawal	23	7 (9.1%)	16 (11.1%)		
Risky behaviors				<0.001 ***	0.615
No	95	65 (83.3%)	30 (21.1%)		
Non-suicidal self-injury (e.g., cutting)	74	11 (14.1%)	63 (44.4%)		
Drug use	7	2 (2.6%)	5 (3.5%)		
Suicidal behaviors	21	0 (0%)	21 (14.8%)		
Non-suicidal self-injury + drugs	1	0 (0%)	1 (0.7%)		
Non-suicidal self-injury + suicidal behaviors	19	0 (0%)	19 (13.4%)		
Non-suicidal self-injury + drugs + suicidal behaviors	3	0 (0%)	3 (2.1%)		
Diagnoses					
Neurodevelopmental disorders ^a^	27	12 (15.6%)	15 (10.3%)	0.347	0.063
Psychotic disorders/symptoms ^b^	47	10 (13%)	37 (25.3%)	0.048 *	0.132
Bipolar disorders	8	3 (3.9%)	5 (3.4%)	1	0
Depressive disorders ^c^	100	15 (19.5%)	85 (58.6%)	<0.001 ***	0.365
Anxiety disorders ^d^	63	14 (18.2%)	49 (33.8%)	0.021 *	0.154
OCD	11	5 (6.5%)	6 (4.1%)	0.656	0.03
PTSD	9	2 (2.6%)	7 (4.8%)	0.722	0.03
Somatic disorder	5	2 (2.6%)	3 (2.1%)	1	0
Eating disorders ^e^	58	20 (26%)	38 (26.2%)	1	0
Sleep disorders ^f^	3	1 (1.3%)	2 (1.4%)	1	0
Gender dysphoria	3	1 (1.3%)	2 (1.4%)	1	0
Conduct disorders ^g^	11	6 (7.8%)	5 (3.4%)	0.274	0.073
Substance use disorder	3	1 (1.3%)	2 (1.4%)	1	0
Personality disorders ^h^	45	4 (5.2%)	41 (28.3%)	<0.001 ***	0.261
Other ^i^	61	25 (32.5%)	36 (24.8%)	0.291	0.071
CGI-S	4.42 ± 1.15	3.63 ± 1.15	4.74 ± 0.99	<0.001 ***	−1.068
CGAS	51.33 ± 14.39	57.36 ± 18.67	48.79 ± 11.31	0.004 **	0.617
Medications at T0	156	48 (61.5%)	108 (74%)	0.076	0.119
Antipsychotics at T0	69	15 (22.1%)	54 (39.1%)	0.022*	0.159
Antidepressants at T0	65	15 (22.1%)	50 (36.2%)	0.052	0.154
Benzodiazepines at T0	76	20 (29.4%)	56 (40.6%)	0.159	0.098
Mood stabilizers at T0	12	1 (1.5%)	11 (8%)	0.109	0.108
Supplements ^j^ at T0	75	23 (33.8%)	52 (37,4%)	0.726	0.024
Medication duration before T0	11.55 ± 17.74	11.94 ± 19.73	11.36 ± 16.77	0.858	0.033
Medications at T1	180	53 (69.7%)	127 (88.8%)	<0.001 ***	0.225
Antipsychotics at T1	93	22 (32.8%)	71 (51.1%)	0.021 *	0.161
Antidepressants at T1	92	25 (37.3%)	67 (47.5%)	0.217	0.086
Benzodiazepines at T1	83	19 (28.4%)	64 (45.4%)	0.028 *	0.152
Mood stabilizers at T1	14	0 (0%)	14 (9.9%)	0.006 **	0.165
Supplements at T1	90	22 (32.9%)	68 (48.3%)	0.052	0.135

Significance: * *p* < 0.05; ** *p* < 0.01; *** *p* < 0.001. Note: ^a^ ADHD, Learning disabilities, Tic; ^b^ Psychosis, Attenuated Psychotic Syndrome, Unspecified Schizophrenia Spectrum and Other Psychotic Disorder, Clinical High Risk for Psychosis; ^c^ Major Depressive Disorder, Dysthymia, Disruptive Mood Dysregulation Disorder; ^d^ Generalized Anxiety Disorder, Separation Anxiety; ^e^ Anorexia Nervosa, Bulimia Nervosa, Binge-eating Disorder, Unspecified Eating Disorders; ^f^ Insomnia, Circadian Rhythm Disorder; ^g^ Oppositive Defiant Disorder, Conduct Disorder; ^h^ Structuring personality disorders—since diagnosing a personality disorder in children and adolescents is not always appropriate, minors who exhibit characteristics of subthreshold personality disorders but do not fully meet the criteria of the DSM-5 are diagnosed with structuring personality disorders; ^i^ Headaches, Migraines; ^j^ Dietary supplements, melatonin, passion flowers.

**Table 2 jcm-14-06412-t002:** Differences between psychotropic prescriptions at T1 [authors’ own processing].

Medication	NSC		SC		
	*n*	%	*n*	%	*p*
Antipsychotics at T1	22	32.8	71	51.1	0.017 *
Antidepressants at T1	25	37.3	67	47.5	0.181
Benzodiazepines at T1	19	28.4	64	45.4	0.023 *
Mood stabilizers at T1	0	0.0	14	9.9	0.007 **
Supplements at T1	22	32.8	68	48.2	0.038 *
Total medications prescribed	53	69.7	127	88.8	0.001 **

Significance: * *p* < 0.05; ** *p* < 0.01.

**Table 3 jcm-14-06412-t003:** Changes in prescriptions at T0 and T1 [authors’ own processing].

Medication	Group	T0 to T1	T1 to T0	χ^2^	*p*
Antidepressants	No concern	12	2	5.786	0.016 *
	Concern	25	8	7.758	0.005 **
Antipsychotics	No concern	12	5	2.118	0.146
	Concern	25	8	7.757	0.005 **
Benzodiazepines	No concern	4	5	0.000	1.000
	Concern	19	11	1.633	0.201
Mood Stabilizers	No concern	0	1	0.000	1.000
	Concern	5	2	0.571	0.445
Supplements	No concern	10	11	0.000	1.000
	Concern	31	15	4.891	0.027 *
Total medications prescribed	No concern	14	9	0.696	0.404
	Concern	28	9	8.757	0.003 **

Significance: * *p* < 0.05; ** *p* < 0.01.

**Table 4 jcm-14-06412-t004:** Logistic linear regressions between pharmacological prescriptions and groups [authors’ own processing].

Medication	OR	CI Lower	CI Upper	*p*
Antipsychotics at T1	2.400	1.089	5.514	0.033 *
Antidepressants at T1	1.112	0.536	2.306	0.774
Benzodiazepines at T1	1.434	0.674	3.082	0.351
Supplements at T1	2.231	1.065	4.806	0.036 *
Total medications prescribed at T1	3.189	1.327	8.007	0.011 *

Significance: * *p* < 0.05. Note: Mood stabilizers were prescribed only in a small subset of patients and exclusively in the SC group. Due to the complete absence of prescriptions of mood stabilizers in the NSC group, logistic regression analyses could not be reliably performed (resulting in a complete separation problem). Therefore, mood stabilizers are reported in Table 2 and Table 3 but excluded from Table 4.

**Table 5 jcm-14-06412-t005:** Changes in prescriptions at T0 and T1 in both risk groups [authors’ own processing].

Medication	Risk	T0 to T1	T1 to T0	χ^2^	*p*
Antidepressants	Low	4	1	0.800	0.371
	High	20	7	5.333	0.021 *
Antipsychotics	Low	7	3	0.900	0.343
	High	18	5	6.261	0.012 *
Benzodiazepines	Low	1	1	0.000	1.000
	High	18	10	1.750	0.186
Mood Stabilizers	Low	0	1	0.000	1.000
	High	5	1	1.500	0.221
Supplements	Low	4	6	0.100	0.752
	High	27	9	8.028	0.005 **
Total medications prescribed	Low	3	2	0.000	1.000
	High	25	7	9.031	0.003 **

Significance: * *p* < 0.05; ** *p* < 0.01.

## Data Availability

The original data presented in the study are openly available in Zenodo at 10.5281/zenodo.16604126 [30].

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
