# Peer review of "Exploring the Link Between Suicidal Concern and Pharmacotherapy in Adolescents: Evidence from a Clinical Cohort"

_jcm, 2025, doi:10.3390/jcm14186412_

Round 1
Reviewer 1 Report
Comments and Suggestions for Authors
The manuscript addresses an important exploring the relationship between suicidal concern and pharmacological prescribing in adolescents. The use of a well-validated instrument (C-SSRS) is a strength. There are several points as follows:
- Definition of “Risky Behaviors”
The term “risky behaviors” appears in Table 1 and the Results section without a clear definition in the Methods. It would be helpful to specify precisely what was included in this category. Since “high-risk” and “low-risk” classifications are based on C-SSRS scores, if “risky behaviors” were also derived from the C-SSRS, then significant differences between these risk groups would be expected by definition. Please clarify the definition for “risky behaviors.” - Medications at T1
Table 1 presents “Medications at T0” but does not provide a corresponding “Medications at T1”. Including this would enhance the reader’s understanding of medication changes over time. - Headaches/Migraines Prevalence
Table 1 indicates 61 patients with “Headaches, Migraines,” which seems high for this sample. Please check the number. In adedition, an analysis restricted to patients with depression (n=100) or depression plus anxiety disorders (n=163) would help determine if the main conclusions remain consistent. - Mood Stabilizers in Table 4
In the Results, it is noted that no significant associations were found for antidepressants, benzodiazepines, or mood stabilizers (with mood stabilizers excluded due to “data issues”). However, in Table 4, mood stabilizers are entirely absent. If data are available, please include them; if not, state explicitly why they were removed. - Conclusions Regarding Benzodiazepines
The first sentence of the Conclusion states:
“This study suggests that suicidal concern significantly influences pharmacological choices in adolescent psychiatric care, often prompting prescriptions aimed at short-term risk containment, particularly antipsychotics and benzodiazepines.”
Given the presented results, it is not clear that an increased use of benzodiazepines can be concluded. Please review whether the conclusion is fully supported by the research findings.
Addressing these points would improve the clarity and interpretability of the results, and ensure that conclusions are firmly grounded in the data presented.
Author Response
Comment 1: The manuscript addresses an important exploring the relationship between suicidal concern and pharmacological prescribing in adolescents. The use of a well-validated instrument (C-SSRS) is a strength.
Response 1: We thank the reviewer for taking the time to read and review the present work and for giving precious suggestions.
Comment 2: There are several points as follows:
- Definition of “Risky Behaviors”
The term “risky behaviors” appears in Table 1 and the Results section without a clear definition in the Methods. It would be helpful to specify precisely what was included in this category. Since “high-risk” and “low-risk” classifications are based on C-SSRS scores, if “risky behaviors” were also derived from the C-SSRS, then significant differences between these risk groups would be expected by definition. Please clarify the definition for “risky behaviors.”
Response 2: We thank the Reviewer for this important comment. We agree that the term “risky behaviors” required clarification. In our study, this category was not based solely on C-SSRS scores, but was derived from multiple sources, including open clinical interviews to collect anamnestic data and the semi-structured interviews K-SADS-PL-DSM-5 and C-SSRS. Specifically, “risky behaviors” included non-suicidal self-injury (e.g., cutting), drug use, suicidal behaviors (as assessed through the C-SSRS), and combinations of these behaviors. We have now clarified this definition in the Participants and Procedures section to better explain the “risky behaviors” reported in Table 1, as well as social relations and school performances.
Comment 3:
- Medications at T1
Table 1 presents “Medications at T0” but does not provide a corresponding “Medications at T1”. Including this would enhance the reader’s understanding of medication changes over time.
Response 3: We really appreciate the Reviewer’s comment. We agree that introducing Medications at T1 is very useful in making the reader more aware of the medication changes between T0 and T1. We introduced it in Table 1 and in Supplementary Table 2 (descriptives of low and high-risk participants from the SC group).
Comment 4:
- Headaches/Migraines Prevalence
Table 1 indicates 61 patients with “Headaches, Migraines,” which seems high for this sample. Please check the number. In adedition, an analysis restricted to patients with depression (n=100) or depression plus anxiety disorders (n=163) would help determine if the main conclusions remain consistent.
Response 4: We appreciate the Reviewer's insightful comment. Regarding the unexpectedly high prevalence of headaches/migraines observed in our sample, this can be explained by the fact that our institution is a referral center for the diagnosis and treatment of pediatric headaches, and several clinical and experimental trials on headaches are ongoing at our hospital. Therefore, a higher-than-average proportion of patients with headache/migraine disorders is expected among those admitted to our unit.
To address potential diagnostic confounding and explore effect heterogeneity, as requested, we repeated the analyses in patients with depression and in those with either depressive or anxiety disorders. Results were directionally consistent with the main analyses. In the depression-only subgroup, differences mostly did not reach statistical significance, likely due to reduced sample size. In contrast, in the depression/anxiety subgroup, higher rates of antipsychotic and overall prescriptions among patients with suicidal concern remained (or became) significant. These sensitivity analyses support the robustness of the main findings. We added them to the paper to provide the reader with more comprehensive results. Tables are reported in Supplementary materials.
Comment 5:
- Mood Stabilizers in Table 4
In the Results, it is noted that no significant associations were found for antidepressants, benzodiazepines, or mood stabilizers (with mood stabilizers excluded due to “data issues”). However, in Table 4, mood stabilizers are entirely absent. If data are available, please include them; if not, state explicitly why they were removed.
Response 5: We thank the Reviewer for noticing this inconsistency. Mood stabilizers were prescribed only in a small minority of cases (n = 14), exclusively within the SC group. As a result, regression analyses could not be reliably performed due to quasi-complete separation. For completeness, we have clarified this point in the Results section and in the note under Table 4, while retaining descriptive and univariate comparisons (Tables 2 and 3).
Comment 6:
- Conclusions Regarding Benzodiazepines
The first sentence of the Conclusion states:
“This study suggests that suicidal concern significantly influences pharmacological choices in adolescent psychiatric care, often prompting prescriptions aimed at short-term risk containment, particularly antipsychotics and benzodiazepines.”
Given the presented results, it is not clear that an increased use of benzodiazepines can be concluded. Please review whether the conclusion is fully supported by the research findings.
Addressing these points would improve the clarity and interpretability of the results, and ensure that conclusions are firmly grounded in the data presented.
Response 6: We thank the Reviewer for highlighting this issue. The Reviewer is right. Our results indicate that SC was associated with a higher likelihood of being prescribed antipsychotics and supplements, as well as an overall greater number of medications at T1. For benzodiazepines, while the SC group received these medications more frequently, the association did not remain statistically significant in the adjusted logistic regression analyses. Therefore, the conclusion has been revised to clearly represent the results.
Reviewer 2 Report
Comments and Suggestions for Authors
This manuscript addresses an important and clinically relevant topic by examining how suicidal concern influences pharmacological treatment decisions in hospitalized adolescents with psychiatric disorders. The study is well-motivated and provides valuable insights into individualized treatment planning in adolescent psychiatry. However, while the findings are noteworthy, there are several methodological and interpretational aspects that warrant further examination and enhancement to ensure a comprehensive understanding of the studied relationships and to bolster the manuscript's overall impact in the field:
Abstract
- In the first paragraph of the Introduction, the authors could strengthen the rationale of the study by emphasizing the seriousness of suicidality, including its prevalence and the adverse outcomes associated with it. Clearly presenting this information would help readers understand why suicidality is an important issue and justify the need for research in this area.
- In several paragraphs of the Introduction, the authors extensively discuss the differences between suicidal ideation and suicide attempts. While this is an important issue, the current study does not categorize participants into ideation-only, attempt-only, both, or neither groups; participants are only divided into SC and NSC groups. To maintain clarity and consistency, I recommend removing the detailed discussion of these differences from the Introduction. Instead, this point could be acknowledged in the Limitations section, noting that future research could examine distinctions among these subgroups.
- The Introduction section needs to be more a critical review of the relevant literature discussing what's out there, what's missing, how the study will fill any gaps in the literature, and how the study would add to the literature.
- In the Introduction, the authors mention that 'Multiple risk factors have been identified over time as contributors to the development of suicidal ideation and behaviors.' While this information is relevant to the broader field, it does not directly address the specific scientific questions of the current study. I suggest that the authors focus the literature review on studies that are directly related to their research questions, clearly identify the research gap, and articulate the necessity and rationale for the present study. This would help ensure that the Introduction effectively frames the scientific problem and supports the study objectives.
- I recommend that the authors provide a more detailed description of the participants’ age, including the mean and standard deviation, as well as the distribution of gender. Including these details would improve the transparency of the sample characteristics and help readers better understand the study population, thereby enhancing the rigor and reproducibility of the research.
- The authors should provide a detailed description of the ethical approval process.The specific approval number.
- Based on this, how many participants were initially included in the study? What was the final attrition rate, and was the attrition random? I recommend that the authors provide a participant inclusion and exclusion flowchart to enhance clarity.
- How did the authors determine the sample size?
- In the Methods section, it appears that the authors have not provided a detailed description of the measurements used in the study. It is important to clarify whether the instruments were culturally adapted for the target population and whether their reliability and validity were evaluated, what are their Cronbach's alpha?Including this information would enhance transparency, strengthen the methodological rigor, and support the replicability of the study.
- Please provide a rationale for the measures used for the study.The description of the reliability and validity of the questionnaire is weak, only the internal consistency reliability is explained, and the validity evidence of the questionnaire tool is lacking.
- The Procedure section should be combined with the Participants section into a single section for better organization and clarity.
- The authors should include a section on covariates in the Assessments part. Additionally, please provide a rationale for the covariates selected.
- Please provide a rationale for the analyses selected. Also, the authors should discuss the missing data and how they were handled.
- In Table 1, there is an error in the notation of the first two p-values. The value 0.019 is greater than 0.01 and should therefore be marked with a single asterisk (*) rather than a double asterisk (**).
- The discussion section is too brief and lacks a comprehensive comparison between the current findings and previous research. The authors need to discuss the implications of their study for future research and practice more deeply, ensuring that these implications are closely aligned with the study’s aims.
Author Response
Comment 1: This manuscript addresses an important and clinically relevant topic by examining how suicidal concern influences pharmacological treatment decisions in hospitalized adolescents with psychiatric disorders. The study is well-motivated and provides valuable insights into individualized treatment planning in adolescent psychiatry. However, while the findings are noteworthy, there are several methodological and interpretational aspects that warrant further examination and enhancement to ensure a comprehensive understanding of the studied relationships and to bolster the manuscript's overall impact in the field:
Response 1: We thank the reviewer for reading and reviewing the present work and for providing suggestions to improve the paper.
Comment 2: Abstract
Response 2: We are unsure if the Reviewer has specific comments related to the Abstract, or if that means the entire abstract needs revision. For completeness, we reviewed and revised the abstract in accordance with the reviewers’ suggestions.
Comment 3:
- In the first paragraph of the Introduction, the authors could strengthen the rationale of the study by emphasizing the seriousness of suicidality, including its prevalence and the adverse outcomes associated with it. Clearly presenting this information would help readers understand why suicidality is an important issue and justify the need for research in this area.
Response 3: We really thank the Reviewer for the comment. We revised the first paragraph of the Introduction, emphasizing and further elaborating on aspects related to suicidal concerns.
Comment 4:
- In several paragraphs of the Introduction, the authors extensively discuss the differences between suicidal ideation and suicide attempts. While this is an important issue, the current study does not categorize participants into ideation-only, attempt-only, both, or neither groups; participants are only divided into SC and NSC groups. To maintain clarity and consistency, I recommend removing the detailed discussion of these differences from the Introduction. Instead, this point could be acknowledged in the Limitations section, noting that future research could examine distinctions among these subgroups.
Response 4: We appreciate the Reviewer's valuable comment and fully agree with this observation. Accordingly, we have removed the detailed paragraph about the differences between suicidal ideation and suicidal attempts from the Introduction section to improve clarity and focus. We also addressed this aspect in the Limitation section, as suggested.
Comment 5:
- The Introduction section needs to be more a critical review of the relevant literature discussing what's out there, what's missing, how the study will fill any gaps in the literature, and how the study would add to the literature.
Response 5: We thank the Reviewer for the comment. We have carefully revised the Introduction section, refining it to maintain a clearer and more focused alignment with the research question addressed in our study.
Comment 6:
- In the Introduction, the authors mention that 'Multiple risk factors have been identified over time as contributors to the development of suicidal ideation and behaviors.' While this information is relevant to the broader field, it does not directly address the specific scientific questions of the current study. I suggest that the authors focus the literature review on studies that are directly related to their research questions, clearly identify the research gap, and articulate the necessity and rationale for the present study. This would help ensure that the Introduction effectively frames the scientific problem and supports the study objectives.
Response 6: We appreciate the Reviewer’s comment. We have revised the Introduction section as suggested, removing information that was not directly relevant to our study.
Comment 7:
- I recommend that the authors provide a more detailed description of the participants’ age, including the mean and standard deviation, as well as the distribution of gender. Including these details would improve the transparency of the sample characteristics and help readers better understand the study population, thereby enhancing the rigor and reproducibility of the research.
Response 7: We appreciated the comment. We preferred not to duplicate the reporting of sociodemographic data in Table 1. However, we have added that information as suggested in the Participant and Procedure section.
Comment 8:
- The authors should provide a detailed description of the ethical approval process.The specific approval number.
Response 8: We added a more in-depth description of the ethical approval process in the Study design section. The approval number was already reported in the “Institutional Review Board Statement” section. For completeness, we also added it to the Study design section.
Comment 9:
- Based on this, how many participants were initially included in the study? What was the final attrition rate, and was the attrition random? I recommend that the authors provide a participant inclusion and exclusion flowchart to enhance clarity.
Response 9: We appreciate the Reviewer's comment. In our study, 350 adolescents were initially considered, of whom 126 were excluded based on predefined criteria (intellectual disability, language proficiency, history of head trauma, or lack of consent), leaving a final sample of 224 participants. As such, the attrition rate was approximately 36%, entirely attributable to a priori exclusion criteria rather than subsequent random drop-outs. This information is now clarified in the manuscript and in the flowchart, included as Figure 1.
Comment 10:
- How did the authors determine the sample size?
Response 10: We thank the Reviewer for this question. As the primary aim of the protocol study was descriptive, no formal sample size calculation was performed. Based on the literature, the expected prevalence ranges from 20.5% in outpatient settings to 52.3% in inpatient settings (Pfeffer et al., 1980, 1982). Over the course of five years, we anticipated enrolling approximately 150 participants, which would have allowed us to estimate a prevalence of around 50% with a precision of ±8% at the 95% confidence level. Nevertheless, the final sample of 224 adolescents exceeded this expectation and provides sufficient statistical power to detect clinically meaningful differences, which we acknowledge as a strength of the study.
Comment 11:
- In the Methods section, it appears that the authors have not provided a detailed description of the measurements used in the study. It is important to clarify whether the instruments were culturally adapted for the target population and whether their reliability and validity were evaluated, what are their Cronbach's alpha?Including this information would enhance transparency, strengthen the methodological rigor, and support the replicability of the study.
Response 11: We thank the Reviewer for this comment. We specify that no self-report questionnaires were administered in the present study. Instead, we relied on semi-structured clinical interviews (K-SADS-PL, SCID-5-PD, C-SSRS) and clinician-rated scales (CGI-S, CGAS), all of which have been validated and widely used in Italian clinical populations. Cronbach’s alpha is not applicable to these measures, as they do not consist of multiple items reflecting a single latent construct (e.g., diagnostic interviews yield categorical diagnoses, while CGI-S and CGAS are global clinician ratings based on a single score). Reliability and validity evidence for these instruments is well established in the literature. We have expanded the Participants and Procedures section to better describe the instruments used and their psychometric properties.
Comment 12:
- Please provide a rationale for the measures used for the study.The description of the reliability and validity of the questionnaire is weak, only the internal consistency reliability is explained, and the validity evidence of the questionnaire tool is lacking.
Response 12: We thank the Reviewer for this observation. As previously indicated, no self-report questionnaires were administered in our study. Instead, we relied on clinician-rated and semi-structured diagnostic interviews, all of which have been extensively validated in the literature. Specifically, suicidal ideation and behaviors were assessed using the C-SSRS, which has demonstrated strong reliability and validity in both clinical and research settings. Psychiatric diagnoses and comorbidities were established using K-SADS-PL-DSM-5 and, for participants aged 14 and older, the SCID-5-PD. Clinical severity and global functioning were assessed using the clinician-report CGI-S and CGAS, both of which are widely used and supported by evidence of reliability and validity. We have revised the manuscript to clarify which type of instruments they are and to better emphasize their psychometric properties.
Comment 13:
- The Procedure section should be combined with the Participants section into a single section for better organization and clarity.
Response 13: We followed the Reviewer’s comment and merged the two sections.
Comment 14:
- The authors should include a section on covariates in the Assessments part. Additionally, please provide a rationale for the covariates selected.
Response 14: We thank the Reviewer for this comment. In the manuscript, we reported that all regression models included age, gender, and psychiatric diagnoses as covariates, as these are well-established factors influencing both suicidality and psychiatric pharmacological prescriptions (“All models were adjusted for age, gender, and psychiatric diagnoses”). We have clarified this rationale in the Data Analysis section in more detail.
Comment 15:
- Please provide a rationale for the analyses selected. Additionally, the authors should discuss the missing data and how it was handled.
Response 15: We thank the Reviewer for this comment. We choose the analyses considering the level of measurement of the variables and the distributional properties of the data. Categorical variables were compared using chi-square or Fisher’s exact tests, the latter being applied when expected cell counts were small. Continuous variables were compared using Welch’s t-test, which does not assume homogeneity of variance and is therefore more robust in clinical samples. Logistic regression was used to examine associations with binary outcomes (pharmacological prescription at T1), allowing adjustment for relevant covariates (age, gender, psychiatric diagnoses). McNemar’s test was chosen to analyze within-subject changes in prescription patterns between T0 and T1, as it is specifically designed for paired categorical data. Finally, polypharmacy was operationalized as the number of prescribed drug classes and compared using Welch’s t-test. Missing data were handled through listwise deletion (complete-case analysis), and no imputation was performed. We clarified our choices regarding statistical analyses in the Data Analysis section.
Comment 16:
- In Table 1, there is an error in the notation of the first two p-values. The value 0.019 is greater than 0.01 and should therefore be marked with a single asterisk (*) rather than a double asterisk (**).
Response 16: We appreciate the reviewer's attention. We updated the asterisk and checked all the tables.
Comment 17:
- The discussion section is too brief and lacks a comprehensive comparison between the current findings and previous research. The authors need to discuss the implications of their study for future research and practice more deeply, ensuring that these implications are closely aligned with the study’s aims.
Response 17: We appreciate the Reviewer’s suggestion. We have carefully reviewed the Discussion and Conclusion sections and made further adjustments to ensure alignment with the study’s aims.
Round 2
Reviewer 1 Report
Comments and Suggestions for Authors
The authors have adequately addressed the previous comments, and the manuscript has been substantially improved. The study design, analyses, and interpretation are appropriate, and the findings are of clear clinical and scientific relevance. I believe the manuscript is now suitable for publication. I recommend acceptance of the manuscript.
Reviewer 2 Report
Comments and Suggestions for Authors
Thank you for your thorough revisions. I believe the manuscript has been greatly improved, and I recommend it for acceptance in its current form.